# Grid-Based Software for Quantification of Diabetic Retinal Nonperfusion on Ultra-Widefield Fluorescein Angiography

**DOI:** 10.3390/diagnostics15070875

**Published:** 2025-03-31

**Authors:** Amro Omari, Caitlyn Cooper, Eric B. Desjarlais, Maverick Cook, Maria Fernanda Abalem, Chris A. Andrews, Katherine Joltikov, Rida M. Khan, Andy Chen, Andrew DeOrio, Thomas W. Gardner, Yannis M. Paulus, K. Thiran Jayasundera

**Affiliations:** 1University of Michigan Medical School, Ann Arbor, MI 48105, USA; amroomari22@gmail.com (A.O.); ridak@umich.edu (R.M.K.); andych@umich.edu (A.C.); 2Department of Ophthalmology and Visual Sciences, W.K. Kellogg Eye Center, University of Michigan, Ann Arbor, MI 48105, USA; cocaitly@med.umich.edu (C.C.); edesjarl@umich.edu (E.B.D.); mabalemd@med.umich.edu (M.F.A.); chrisaa@med.umich.edu (C.A.A.); katherinejoltikov@gmail.com (K.J.); tomwgard@med.umich.edu (T.W.G.); 3Department of Electrical Engineering and Computer Science, University of Michigan, Ann Arbor, MI 48109, USA; loopwpdf8@mozmail.com (M.C.); awdeorio@umich.edu (A.D.); 4Department of Biomedical Engineering, University of Michigan, Ann Arbor, MI 48109, USA; 5Wilmer Eye Institute, Johns Hopkins University, Baltimore, MD 21287, USA

**Keywords:** fluorescein angiography, diabetic retinopathy, retinal hypoperfusion, retinal nonperfusion, ultra-widefield imaging

## Abstract

**Background/Objectives:** Fluorescein angiography (FA) is essential for diagnosing and managing diabetic retinopathy (DR) and other retinal vascular diseases and has recently demonstrated potential as a quantitative tool for disease staging. The advent of ultra-widefield (UWF) FA, allowing visualization of the peripheral retina, enhances this potential. Retinal hypoperfusion is a critical risk factor for proliferative DR, yet quantifying it reliably remains a challenge. **Methods:** This study evaluates the efficacy of the Michigan grid method, a software-based grading system, in detecting retinal hypoperfusion compared to the traditional freehand method. Retinal UWF fluorescein angiograms were obtained from 50 patients, including 10 with healthy retinae and 40 with non-proliferative DR. Two independent, masked graders quantified hypoperfusion in each image using two methods: freehand annotation and a new Michigan grid method. **Results:** Using the Michigan grid method, Grader 1 identified more ungradable segments, while Grader 2 identified more perfused and nonperfused segments. Cohen’s weighted kappa indicated substantial agreement, which was slightly higher for the entire retina (0.711) compared to the central retinal area (0.686). The Michigan grid method shows comparable or slightly improved inter-rater reliability compared to the freehand method. **Conclusions:** This study demonstrates a new Michigan grid method for the evaluation of FA for hypoperfusion while highlighting ongoing challenges in achieving consistent and objective retinal nonperfusion assessment, underscoring the need for further refinement and the potential integration of automated approaches.

## 1. Introduction

Fluorescein angiography is an important part of the diagnosis and management of retinal vascular diseases. However, there is some promise that it can become a quantitative clinical measure to describe disease severity and stratify risk in patients with diabetic retinopathy (DR). In 1991, the Early Treatment Diabetic Retinopathy Study (ETDRS) identified several fluorescein angiography risk factors for progression to proliferative diabetic retinopathy that may not have been appreciable on clinical examination or color imaging [1]. These risk factors included fluorescein leakage, capillary loss, capillary dilation, and retinal hypoperfusion, and were observed from a series of seven overlapping photographs capturing nearly 75 degrees of retinal area [1]. The more recent development of ultra-widefield imaging has allowed for the instantaneous imaging of 200 degrees, or approximately 82%, of the retinal surface. Studies using ultra-widefield fundus photography have revealed a correlation between peripheral lesions outside of the ETDRS area and increased risk of progression to proliferative diabetic retinopathy [2,3]. In their 2012 study of 236 eyes belonging to patients with diabetes, Matthew Wessel et al. demonstrated the ability of ultra-widefield fluorescein angiography to image 3.2 times more retinal surface and 3.9 times more retinal nonperfusion than the ETDRS methodology [4]. Numerous other studies have evaluated the extent and distribution of hypoperfusion in the eyes of patients with diabetes; however, the interpretation of these studies and the clinical utility of ultra-widefield fluorescein angiography is limited by the current methods of quantifying retinal hypoperfusion.

Substantial effort has been made in recent years to develop methods for the quantitative evaluation of ischemic changes in ultra-widefield fluorescein angiography. Despite these efforts, there has yet to emerge a method or technology with the efficiency, reliability, and objectivity required for clinical implementation. The most commonly used approach at present is the ischemic index method. This method is labor intensive, requiring the grader to manually outline areas of retinal nonperfusion on each fluorescein angiogram [5,6]. Nonperfusion is reported as the ratio of nonperfused area to that of the entire visible retina. Nonperfusion exceeding a certain threshold can then predict the risk of proliferative diabetic retinopathy [7]. Grid-based approaches have also been developed, which require graders to evaluate nonperfusion in discreet image segments delineated by a grid overlaid on the angiogram. Unlike the ischemic index method, this method preserves information regarding the topographic distribution of nonperfusion [8].

Both approaches, however, are subject to graders’ individual ability to recognize hypoperfusion. Computerized methods have been developed that either automate the identification of hypoperfusion or nonperfusion [9,10,11,12,13], correct for the projection of the three-dimensional retina on a two-dimensional plane [14,15], or both. Studies have also used concentric circle grids to quantify the topographic distribution of their automatic nonperfusion grading [16]; however, these methods often fail to distinguish physiologic from pathologic hypoperfusion. In our current study, we present software that confronts these limitations by standardizing fluorescein angiography images, segmenting each image according to a grid of concentric rings, grading each image segment in comparison to a healthy control, and creating a framework that may be more easily translated to an artificial intelligence platform. The purpose of this study is to compare our software-based grading system to standard freehand grading of Optos images. Primary measures include the ability of both grading systems to measure surface area of nonperfusion and intergrader agreement amongst graders.

## 2. Materials and Methods

This retrospective cross-sectional study was performed at the Kellogg Eye Center, University of Michigan in Ann Arbor, MI, USA. This is a tertiary care academic medical center whose retina clinic treats patients with advanced retinal diseases. The study was approved by the University of Michigan Institutional Review Board and conducted according to the Declaration of Helsinki.

### 2.1. Patient Selection

We included forty patients with diabetic retinopathy in different stages and ten patients with no evidence of retinal pathology. All patients had undergone retinal ultra-widefield fluorescein angiography (UWF FA) using the Optos P200Tx (Optos PLC, Dunfermline, Scotland, UK) following complete vessel saturation during the fluorescein late arteriovenous phase (between 40 and 90 s) to allow visualization of the retinal vasculature and regions lacking perfusion. Patients with evidence of retinal disease other than diabetic retinopathy, media opacity, or both were not included. Only one eye was included per patient.

### 2.2. Grading Protocol

Two graders were selected for the assessment of ultra-widefield fluorescein angiograms. Grader 1 (A.O.) and Grader 2 (M.F.A.) were trained for the study protocol prior to grading. Both graders quantified hypoperfusion in each of the 50 fluorescein angiography images using two different protocols, the freehand method and the Michigan Method, each described below.

### 2.3. Freehand Method

Each of the 50 two-dimensional fluorescein angiography images were projected onto a three-dimensional sphere with a diameter of 24 mm using proprietary software available from Optos as described previously (Optos ProView, Registration and Area Measurement Software, online upload, accessed 2017 to 2020, Dumfermline, Scotland).^23^ Areas of hypoperfusion were then outlined by each grader using image segmenting software (ITK-SNAP, Version 3, www.itksnap.org, Figure 1). The segmented images were then exported to a second type of Optos software, which calculated the areas of outlined hypoperfusion in square millimeters.

### 2.4. Michigan Grid Method

Each of the 50 fluorescein angiography images were uploaded to our online hypoperfusion grading software, created for the purposes of this study. For each image, the software prompted the user to identify and click on the fovea and the center of the optic disc. A grid of 289 cells arranged in 8 concentric rings around a central circular cell was overlaid on top of the image. The software shifted and rotated the image to center the fovea on the grid and align the fovea and the optic disc on the grid’s horizontal axis. The grid was scaled such that the distance between the fovea and the center of the optic disc spanned 9.5% of the grid’s diameter (Figure 2). The Python (Version 3, Python Software Foundation, Wilmington, DE, USA) code segments used to rotate, scale, and shift the fluorescein angiogram and grid are shown in Figure 3 and the full code is freely available online at https://github.com/OpenHealthSoftware/umich-isla-prototype/ (accessed on 10 March 2025). By assuming that the distance between the fovea and the optic disc was constant between eyes, the dimensions of the grid cells were determined such that each image segment contained an equivalent area of the fluorescein angiogram across different eyes (Figure 4). The distance between the fovea and the optic disc was measured to be 4.76 mm with a standard deviation of 0.34 mm in a previous study of 2836 individuals using fundus photography [17]. The method used to divide the disc into cells of equal area was described in 2012 by Benoit Beckers and Pierre Beckers [18]. The Michigan grid method involved the manual grading of each of these 289 cells but allowed for the localization of the grading of nonperfusion or hypoperfusion to a specific, defined spatial region for further segmentation of the image.

The user then graded each of the 289 cells. Cells to be graded were presented individually to the user alongside a control image that corresponded with the same area of retina in a healthy eye. The user graded each cell as either perfused, 76–99% perfused, 51–75% perfused, 26–50% perfused, 1–25% perfused, or nonperfused. For descriptive statistics and comparison between graders, cells were assigned a perfusion score of 1, 0.875, 0.625, 0.375, 0.125, or 0, respectively. The perfusion of an area of retina was defined as the average perfusion score of the cells within that area, measured in percent perfusion and converted for data analysis to percent hypoperfusion. For categorical comparisons, cells were identified as perfused, hypoperfused (51–99% perfused), or nonperfused (0–50% perfused). Cells were considered ungradable if they did not have at least one visible vessel within the cell area or if there was evidence of an image artifact.

Hypoperfusion of three retinal areas was considered for the purposes of this study: the entire fluorescein angiogram, a central area of 121 cells corresponding with the ETDRS’s seven standard fields [19] of the retina, and the peripheral area outside of the central ETDRS area.

### 2.5. Statistical Analysis

Continuous variables were summarized by means, standard deviation, and Tukey’s five-number summary. Categorical variables were summarized by counts and percentages. Continuous scores from two graders were compared using Bland–Altman plots and Kendall’s tau coefficient. Categorical scores from the two graders were compared using cross-tabulation and Cohen’s (weighted) kappa coefficient. Image-level bootstrapping was used to compute 95% confidence intervals. Analysis was performed using R (R Version 4.4.3, The R Foundation for Statistical Computing, Vienna, Austria).

## 3. Results

Ultra-widefield fluorescein angiograms of one representative eye from each of 50 patients were graded to quantify retinal nonperfusion. Of the 50 angiograms, 10 represented healthy retinae, and the remaining 40 were affected by varying degrees of non-proliferative diabetic retinopathy with no other known retinal pathology. Nonperfusion was quantified in each of the 50 angiograms by two graders using both the Michigan Method and the freehand method.

Across the 10 healthy retinae, Grader 1 measured an average of 0.0% nonperfusion and Grader 2 measured an average of 0.1% nonperfusion using the Michigan Method. With the freehand method, Grader 1 measured an average of 0.2 mm^2^ of nonperfusion and Grader 2 measured an average of 1.7 mm^2^ of nonperfusion. The descriptive statistics used for the quantification of hypoperfusion across the 40 pathologic retinae are listed in Table 1. Grader 1 quantified less hypoperfusion than Grader 2 using both the Michigan Method and the freehand method.

### 3.1. Michigan Method Grading of the Entire Retina

To quantify nonperfusion across the entire visible retina captured in each of the 50 ultra-widefield fluorescein angiograms, each grader analyzed a total of 11,900 image segments using the Michigan Method. The percentages of cells identified as perfused, hypoperfused, nonperfused, or ungradable by each grader are listed in Table 2. Of the 11,900 segments, 33% were assigned the same score by each grader, including ungradable segments. Grader 1 identified 4760 (40% of the total) more cells as ungradable relative to Grader 2. Grader 2 identified 4165 (35% of the total) more cells as perfused and 833 (7%) more cells as nonperfused compared to Grader 1, whereas Grader 1 identified 238 (2%) more cells as hypoperfused compared to Grader 2.

A total of 3251 image segments remained after disregarding the 8649 (73%) segments that were marked “ungradable” by either grader. The percentages of cells identified as perfused, hypoperfused, or nonperfused after disregarding the cells marked as ungradable by either grader are listed in Table 3. Grader 2 identified 98 (3%) more cells as perfused and 260 (8%) more cells as nonperfused compared to Grader 1. Grader 1 identified 325 (10%) more cells as hypoperfused compared to Grader 2.

### 3.2. Michigan Method Grading of the Central and Peripheral Retina

Percent nonperfusion was recalculated for each of the 50 ultra-widefield fluorescein angiograms, considering the central area of retina corresponding with the area contained within the ETDRS’s seven standard fields [16]. A total of 4900 image segments fell within this area, and the percentages identified as perfused, hypoperfused, nonperfused, or ungradable are listed in Table 2. Of the 4900 total segments, 50% were assigned the same score by each grader, including ungradable segments. Grading trends were similar compared to the Michigan Method applied to the entire retina. Grader 1 identified 1911 (39%) more cells as ungradable compared to Grader 2. Grader 2 identified 1715 (35%) more cells as perfused and 490 (10%) more cells as nonperfused compared to Grader 1. Grader 1 identified 294 (6%) more cells as hypoperfused compared to Grader 2. Of note, Grader 1 identified 92% of the 7000 image segments outside of the central ETDRS area of the retina as ungradable, whereas Grader 2 identified 53% of these peripheral image segments as ungradable. Grader 1 and Grader 2 identified 5% and 40% of these peripheral image segments as perfused, respectively (Table 2).

A total of 2712 image segments remained after disregarding the 2188 (45%) segments that were marked “ungradable” by either grader. The percentages of cells identified as perfused, hypoperfused, of nonperfused by each grader are listed in Table 3. Grader 2 identified 108 (4%) more cells as perfused and 190 (7%) more cells as nonperfused compared to Grader 1. Grader 1 identified 298 (11%) more cells as hypoperfused compared to Grader 2.

### 3.3. The Inter-Rater Reliability of the Freehand Versus the Michigan Method

Inter-rater reliability was assessed for the freehand method and the Michigan Method, considering the entire visible retina, using Bland–Altman plots and rank-based Kendall’s tau coefficient. For the freehand method, Bland–Altman analysis demonstrated strong differences between the graders (Figure 5). The area of nonperfusion was much larger when measured by Grader 2 compared to Grader 1. This difference became more pronounced at higher levels of nonperfusion averaged between the two graders. The rank-based Kendall’s tau coefficient measured 0.686.

Bland–Altman analysis of the Michigan Method, considering the entire retina, also demonstrated strong differences between the graders (Figure 6). Percent nonperfusion was much greater when measured by Grader 2 compared to Grader 1, especially at higher levels of nonperfusion averaged between the two graders. The rank-based Kendall’s tau coefficient measured 0.761.

### 3.4. The Inter-Rater Reliability of the Michigan Method, Central Versus Entire Retina

Inter-rater reliability was further assessed for the Michigan Method, considering only the ETDRS 7SF area of the retina, and compared to the Michigan Method, considering the entire visible retina. The Cohen’s weighted kappa coefficient was compared between the two areas. Across the entire retina, Cohen’s weighted kappa measured 0.711 (95% CI 0.646–0.754). Across the central ETDRS image segments, Cohen’s weighted kappa measured 0.686 (95% CI 0.614–0.742).

## 4. Conclusions

In this study, we created software for the quantification of retinal nonperfusion that builds upon grid-based analysis methods. Currently, the most used method to quantify nonperfusion is the ischemic index. This method, which requires graders to manually outline areas of nonperfusion, depends on the grader’s memory of a healthy retina and ability to precisely delineate areas of pathologic nonperfusion [5,6]. Other approaches have been developed that superimpose grids onto fluorescein angiograms. Grid-based methods allow researchers to investigate the spatial distribution of retinal nonperfusion by subdividing the angiogram and require graders to analyze cells individually. These approaches preserve information regarding the topographic distribution of nonperfusion and reduce the ambiguity introduced by freehand analysis [8,16,20,21,22]. Nevertheless, these grid-based methods depend on the grader’s ability to recall the appearance of a healthy retina. The Michigan Method aims to improve the objectivity of ultra-widefield fluorescein angiogram analysis by standardizing the image’s presentation to the grader, dividing the angiogram into discreet segments, and providing the grader with a direct comparison to an image segment containing an equivalent area of healthy retina when evaluating hypoperfusion.

We compared the ability of two graders to quantify nonperfusion in a series of ultra-widefield fluorescein angiography images affected by various degrees of diabetic retinopathy using the Michigan Method and the freehand method, which utilizes software developed by Optos to measure the precise area of nonperfusion outlined within a hand-drawn area. We hypothesized that the implementation of a grid-based approach with direct comparison to a healthy control would improve inter-rater reliability relative to the freehand method by reducing dependence upon each grader’s individual ability to recognize pathologic hypoperfusion. Overall, we observed results produced through the Michigan Method that were similar or modestly improved relative to those of the freehand method, measured by Bland–Altman analysis and Kendall’s tau coefficient. However, there remained substantial disagreements between graders, especially with respect to the identification of ungradable image segments, which is a major limitation of this study. The manuscript would be significantly strengthened by more rigorous training and standardization of both graders prior to initiation of the study, with a set of test images to ensure that the graders were well-acquainted with the grading criteria, and with a third grader to adjudicate any discrepancies between graders.

We thought that this lack of improvement may have been attributed to peripheral defects of ultra-widefield fluorescein angiography. Peripheral distortion related to the projection of the three-dimensional retina on a two-dimensional surface is well documented [15,23,24]. Although the freehand method, which corrects peripheral distortion, was the basis of comparison for our study, our proof-of concept software does not currently correct such effects. Furthermore, the peripheral retina contains artifacts, such as eyelash shadows and poor illumination, which may interfere with grading and contribute to errors when determining whether an image segment is gradable. For these reasons, we compared the performance of the two graders as they used the Michigan Method to analyze the entire retina, the central ETDRS seven standard fields [19] area of retina, and the peripheral retina outside of the ETDRS area.

In our investigation of grader performance using the Michigan Method for different areas of retina, Grader 1 consistently marked more image segments as ungradable compared to Grader 2 across all three areas. The difference between the percentage of total image segments considered ungradable by Grader 1 compared to Grader 2 was 40 points across the entire retina, 39 points in the central ETDRS area, and 39 points in the peripheral area (Table 2). Although substantially more image segments were considered ungradable by both graders in the peripheral compared to the central retina, the difference between the total number of cells considered ungradable by each grader remained relatively constant. Furthermore, Cohen’s kappa coefficient of intra-rater reliability decreased when considering only the central area of retina. These findings indicate that image artifacts measured as ungradable image segments were more prevalent in the peripheral retina, but the detrimental effect of image artifacts did not entirely account for the limited improvement in inter-rater reliability relative to the freehand method.

To compare the graders’ evaluation of hypoperfused image segments using the Michigan Method, we repeated the analysis while disregarding image segments that were considered ungradable by either grader. After doing so, Grader 1 and Grader 2 identified a similar number of perfused image segments, differing by less than five percentage points across each area of retina; however, there were large differences regarding the extent of hypoperfusion identified in abnormal image segments (Table 3). Across the entire retina, Grader 1 marked twice as many image segments as hypoperfused than nonperfused, while Grader 2 marked twice as many image segments as nonperfused than hypoperfused. These findings suggest that, even with direct comparison to a healthy control and strict criteria guiding the identification of ungradable image segments, determining the extent of hypoperfusion using current methods of fluorescein angiography analysis depends strongly upon grader training and clinical experience.

A limitation of our study is the involvement of Grader 1, a medical student at the time, who lacked sufficient clinical experience in FA image analysis and was not sufficiently trained prior to performing the grading. Future studies may benefit from excluding medical students as graders and including a third grader, ideally a retina specialist, to reliably assess intergrader agreement and adjudicate in cases of discrepancies between graders. Alternatively, artificial intelligence (AI) or other automated systems for grading could be utilized. The significant differences in ungradable segments between Grader 1 and Grader 2 (53% to 92% in the peripheral area and 45% to 73% in the ETDRS area) underscore the necessity of a third grader and improved grader training and standardization to mediate discrepancies.

The disagreements between graders in our study may be attributable to the lack of a clear definition of pathologic hypoperfusion. By including direct comparison to a healthy control in our grading, we hoped to overcome this barrier; however, marked differences persisted. There currently exist no standard criteria in the literature for the identification of nonperfusion on fluorescein angiography. Hypofluorescence in fluorescein angiography is a proxy for pathologic nonperfusion, and the background features must be understood to determine what is abnormal. Current categorization depends on the grader’s clinical experience, with disagreements existing even between retina specialists. A standardized, non-biased assessment must be established to develop a clinically useful measure of retinal nonperfusion.

The design of our software resembles the grid-based concentric rings approach defined by Nicholson et al. [8]. In their study, the performance of the concentric rings approach was compared to the ischemic index method by asking five graders to analyze 28 images using each method. Like our study, they found the intraclass correlation coefficient for the ratio of perfused to nonperfused retina to be similar (difference < 0.05) between the two methods. They also found that inter-rater reliability decreased with increasing distance from the fovea. This finding contrasts with our study, which revealed a slight decrease when comparing the inter-rater reliability of the central ETDRS retinal area versus the entire retinal area. This may be explained by the algorithm used to create the grid in the Michigan Method. Whereas the area of each grid cell increases with distance from the fovea in Nicholson’s concentric rings approach, the area of each grid cell is the same regardless of position when using the Michigan Method. Variability may be introduced by attempts to summarize hypoperfusion over larger areas; thus, standardizing the area of grid cells in our software seems to reduce spatial differences in inter-rater reliability.

Methods to automate the detection of retinal nonperfusion are currently being developed. Texture segmentation, homomorphic filtering, and other methods have emerged from the field of computer science to potentially address the problem of objectively identifying retinal nonperfusion [9,10,11]. One approach was recently published by a group which automatically identifies areas of hypoperfusion by detecting pixels in ultra-widefield angiograms that fall beneath a certain luminosity threshold, outlining darker areas and quantifying nonperfusion in a similar fashion to the ischemic index method [12]. Another approach uses automated grading software with a six-ring concentric circular grid, similar to the one Nicholson et al. [8] used, with the grid areas increasing as they move away from the fovea [16]. Although these approaches show promise, they are currently limited by their inability to distinguish pathologic from physiologic nonperfusion. Segmenting angiograms using a grid and directly comparing each segment to a healthy control, as described in our study, may provide a solution for this problem. For instance, setting the pixel luminosity threshold on a per-image segment basis relative to a healthy eye may prevent regions like the foveal avascular zone from being incorrectly identified as pathologically nonperfused.

We plan to augment our software with the ability to identify and quantify vascular abnormalities such as capillary loss, capillary dilation, microaneurysms, neovascularization, and vascular pruning on a per-image segment basis, with correction for distortion caused by the projection of the retina onto a two-dimensional plane. The presence of such lesions in the retinal periphery, outside of the ETDRS’s seven standard fields, has been associated with diabetic retinopathy severity and retinal nonperfusion [2,3,25]. By standardizing and segmenting fluorescein angiograms according to a grid, they are prepared in a format that is amenable to machine learning. Assessment by machine learning requires a structured input, such as predefined subdivisions of a fluorescein angiogram. After a training period in which image features are “taught” to the algorithm by manual labeling, the algorithm is then able to identify the presence of the same features in subsequent images [26]. Although a standardized approach for the identification of pathologic hypoperfusion has yet to be developed, machine learning may prove to be a useful asset that considers fluorescence patterns in the context of background angiography features more reliably than is currently possible with human graders.

## Figures and Tables

**Figure 1 diagnostics-15-00875-f001:**
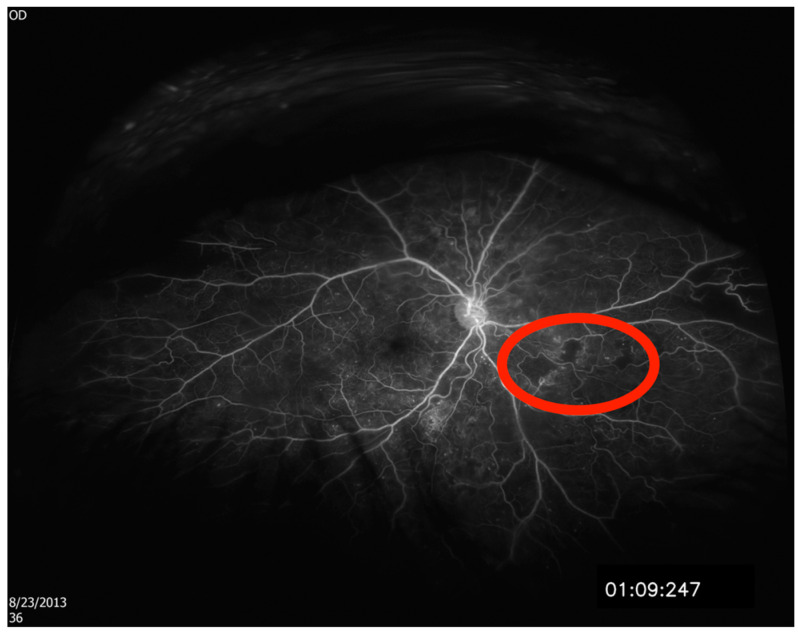
An example of a manually delineated area of retinal nonperfusion used to determine the area of nonperfusion by the freehand method. Red circle demonstrates areas of nonperfusion (black) adjacent to blood vessels.

**Figure 2 diagnostics-15-00875-f002:**
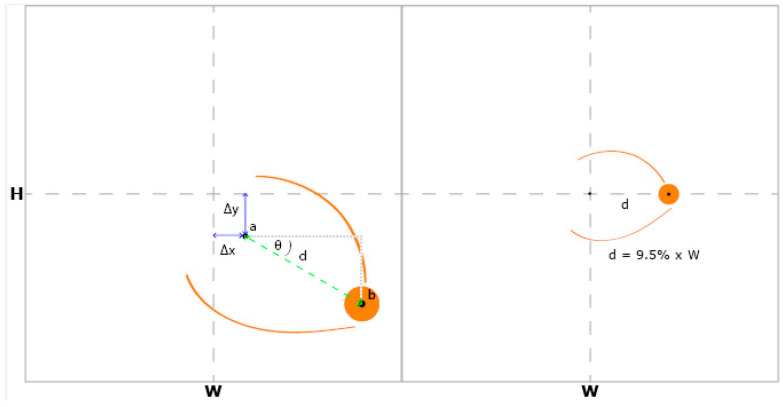
The fluorescein angiogram is shifted by (−Δx, −Δy) to align the fovea (a) with the center of the grid. The image is rotated by θ to align the fovea (a) and the center of the optic disc (b) with the grid’s horizontal axis. The grid is scaled by 9.5/d so that the distance (d) between the fovea (a) and the center of the optic disc (b) is 9.5% of the grid’s width.

**Figure 3 diagnostics-15-00875-f003:**
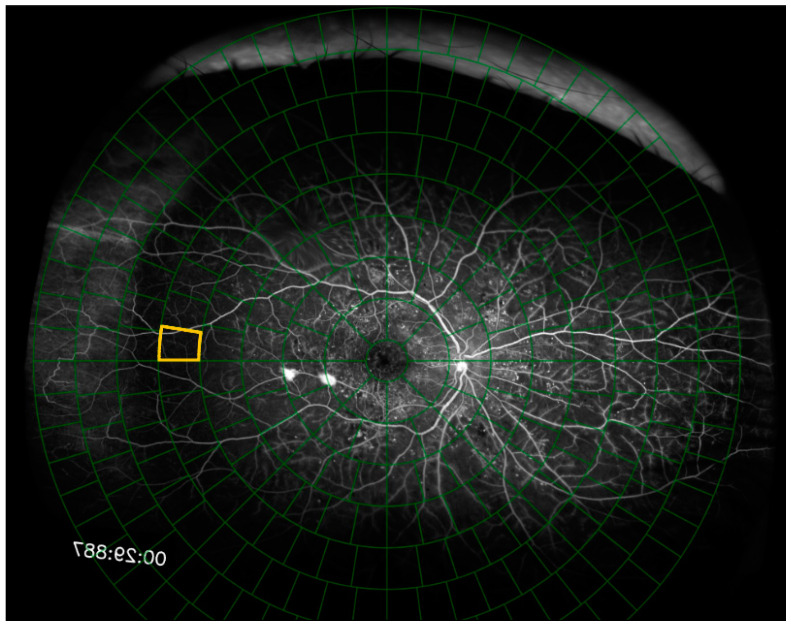
The fluorescein angiogram is presented to the user with the grid overlaid on the fluorescein angiogram. Image segments are presented to the user individually, and the currently selected image segment is outlined in yellow.

**Figure 4 diagnostics-15-00875-f004:**
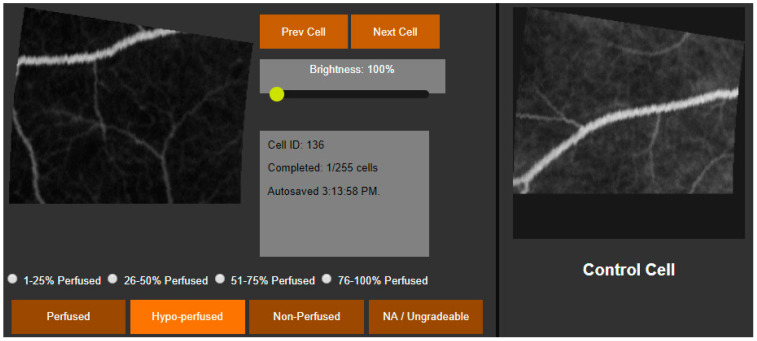
The grader marks each image segment as perfused, one of 4 levels of hypo-perfused, nonperfused, or ungradable by direct comparison to a control cell. The brightness slider adjusts the brightness of the assessment cell and the control cell simultaneously to help visualize image features while maintaining an accurate comparison.

**Figure 5 diagnostics-15-00875-f005:**
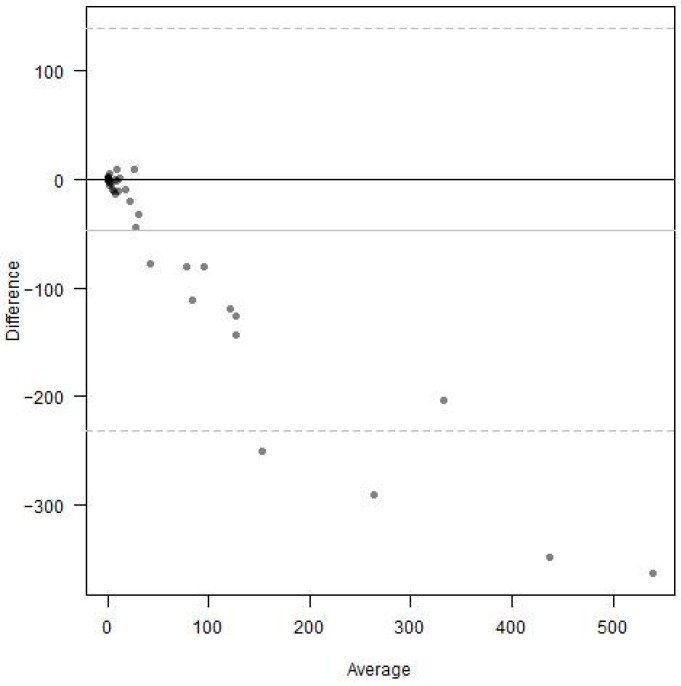
Bland–Altman analysis comparing the grading performance of Grader 1 and Grader 2 using the freehand method.

**Figure 6 diagnostics-15-00875-f006:**
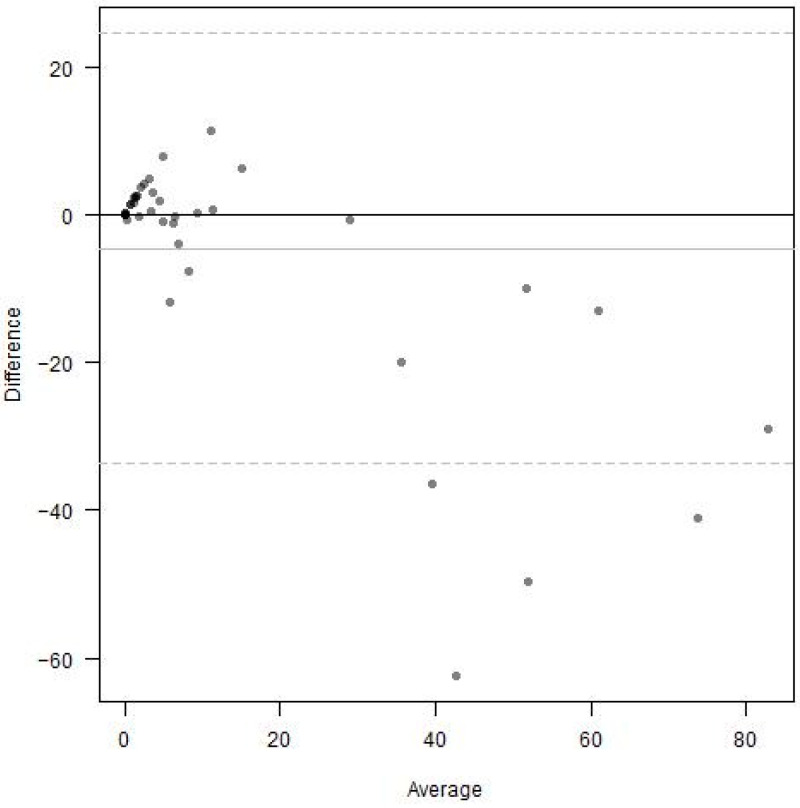
Bland–Altman analysis comparing the grading performance of Grader 1 and Grader 2 using the Michigan Method.

**Table 1 diagnostics-15-00875-t001:** The descriptive statistics used for the grading of ultra-widefield fluorescein angiograms by the Michigan Method and the freehand method.

	Michigan Method		Freehand Method	
	Grader 1	Grader 2	Grader 1	Grader 2
Average	11.9%	17.6%	36.2 mm^2^	93.9 mm^2^
Minimum	0.0%	0.0%	0.0 mm^2^	0.0 mm^2^
Maximum	68.2%	97.3%	357.4 mm^2^	720.6 mm^2^
Median	4.8%	3.4%	5.9 mm^2^	11.4 mm^2^

**Table 2 diagnostics-15-00875-t002:** A comparison of the grading performance between Grader 1 and Grader 2’s assessment of the entire retina, the peripheral retina, and the central ETDRS 7SF area of retina, including any image segment that was marked “ungradable” by either grader.

	Michigan Method—All Image Segments					
	Grader 1			Grader 2		
	Entire Retina	Peripheral Area	ETDRS Area	Entire Retina	Peripheral Area	ETDRS Area
Cells Graded	11,900	7000	4900	11,900	7000	4900
Perfused	20%	5%	40%	55%	40%	75%
Hypoperfused	5%	1%	11%	3%	1%	5%
Nonperfused	2%	1%	4%	9%	6%	14%
Ungradable	73%	92%	45%	33%	53%	6%

**Table 3 diagnostics-15-00875-t003:** A comparison of the grading performance between Grader 1 and Grader 2’s assessment of the entire retina, the peripheral retina, and the central ETDRS 7SF area of retina, disregarding any image segment that was marked “ungradable” by either grader.

	Michigan Method—Disregarding “Ungradable”					
	Grader 1			Grader 2		
	Entire Retina	Peripheral Area	ETDRS Area	Entire Retina	Peripheral Area	ETDRS Area
Cells Graded	3251	539	2712	3251	539	2712
Perfused	72%	70%	72%	75%	70%	76%
Hypoperfused	19%	16%	20%	8%	6%	9%
Nonperfused	9%	14%	8%	17%	24%	15%

## Data Availability

Full python code used to rotate, scale, and shift the fluorescein angiogram and grid is freely available online at https://github.com/OpenHealthSoftware/umich-isla-prototype/. De-identified data will be provided upon reasonable request to the corresponding author in accordance with the law and with the IRB requirements. Due to patient privacy and ethical restrictions (HIPAA), no identifiable patient information will be provided.

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
