# Peer review of "Grid-Based Software for Quantification of Diabetic Retinal Nonperfusion on Ultra-Widefield Fluorescein Angiography"

_diagnostics, 2025, doi:10.3390/diagnostics15070875_

Round 1
Reviewer 1 Report
Comments and Suggestions for Authors
Dear Author,
The manuscript has been written very neat and scientific, especially discussion part is very attractive and holistic. I adore your efforts for preparing such research. But there is a great issue for your paper that is inconclusive and couldn’t add much to our previous results.
There are some huge different between two image graders that you have been chosen and this might result in great bias and inconclusiveness in your study. More than 50 precents of images has been graded as “ungradable”. Although you addressed your weak points in your study but it doesn’t support your manuscript enough for being published. I think you may repeat this method with collaboration of more image grader to get more conclusive results.
You also used the Optus for wide field fluorescein angiography. I think due to widefield thick lens that incorporated in this device, the illumination uptake and image quality of the output images is not enough for your new grading software. I kindly advise you to use another widefield apparatus. Here are some points that might be help you understanding my point of view.
1. Line 136: Is figure 5 addressing correct here?
2. Table 1 revealed that there is significant difference between two image graders. It decreases the power of the study.
3. Regarding table 2, more than 50 percent of images area in total retina and peripheral retina are ungradable and this could affect results erroneously.
With best.
Reviewer 2 Report
Comments and Suggestions for Authors
What is the major utility of using this algorithm vs manual method, as it also depends on human assessment only in smaller segmented areas?
Reviewer 3 Report
Comments and Suggestions for Authors
The article titled "Grid-based Software for Quantification of Diabetic Retinal Nonperfusion on Ultra-widefield Fluorescein Angiography" is well structured in general terms, and they evaluated the role of the Michigan method in calculating the ischemic index with free hand marking and compared the sensitivity of measurements made with both methods.
Major points
"1) The article is being evaluated by a general medical journal, Diagnostics, and the methodology should be explained and detailed so that the general reader can understand more.
2) The online site where the Michigan method is used should be clearly stated with a link. (3rd page, lines 110–111)
3) The paragraph comparing the study conducted by Nicholson in the discussion is not convincing enough. (10th page, lines 344–356) Regardless, more variation is expected to be seen, considering that it may be affected by peripheral retinal artifacts and more ischemic changes in diabetic retinopathy. However, the results were paradoxically different. A more convincing discussion in this respect would contribute to the article. However, how clinically significant is the difference between the values ​​in the conclusion section? The authors can also bring a different perspective to this issue.
"Cohen's weighted kappa measured 0.711 (95% CI 0.646–0.754). Across the central ETDRS image segments, Cohen's weighted kappa measured 0.686 (95% CI 0.614-0.742). "
Minor points
1) The numbers in the following sentence should be checked.
"The significant differences in ungradable segments between Grader 1 and Grader 2 (53% to 92% in the peripheral area and 45% to 73% in the ETDRS area) underscore the necessity of a third grader to mediate discrepancies." (page 10, lines 331-332)
2) There are errors in the references.
6. Author name and journal name
9. The journal name is incorrect
15. The journal name is missing
23. The journal name is incorrect (Retina, not Retin)
All references should be checked with a program like Endnote or Mendeley.
Round 2
Reviewer 3 Report
Comments and Suggestions for Authors
Thanks for implementation of my suggestion.